# Proteomic Analysis Comparison on the Ecological Adaptability of Quinclorac-Resistant *Echinochloa crus-galli*

**DOI:** 10.3390/plants12040696

**Published:** 2023-02-04

**Authors:** Lamei Wu, Can Wu, Haona Yang, Jiangshan Yang, Lifeng Wang, Shangfeng Zhou

**Affiliations:** 1Hunan Weed Science Key Laboratory, Hunan Academy of Agricultural Sciences, Changsha 410125, China; 2Hunan Agricultural Biotechnology Research Institute, Hunan Academy of Agricultural Sciences, Changsha 410125, China

**Keywords:** quinclorac-resistant *E. crus-galli*, quantitative proteomics, photosynthetic parameters, GST activity, ecological adaptability

## Abstract

Barnyardgrass (*Echinochloa crus-galli* L.) is the most serious weed threatening rice production, and its effects are aggravated by resistance to the quinclorac herbicide in the Chinese rice fields. This study conducted a comparative proteomic characterization of the quinclorac-treated and non-treated resistant and susceptible *E. crus-galli* using isobaric tags for relative and absolute quantification (iTRAQ). The results indicated that the quinclorac-resistant *E. crus-galli* had weaker photosynthesis and a weaker capacity to mitigate abiotic stress, which suggested its lower environmental adaptability. Quinclorac treatment significantly increased the number and expression of the photosynthesis-related proteins in the resistant *E. crus-galli* and elevated its photosynthetic parameters, indicating a higher photosynthetic rate compared to those of the susceptible *E. crus-galli*. The improved adaptability of the resistant *E. crus-galli* to quinclorac stress could be attributed to the observed up-regulated expression of eight herbicide resistance-related proteins and the down-regulation of two proteins associated with abscisic acid biosynthesis. In addition, high photosynthetic parameters and low glutathione thiotransferase (GST) activity were observed in the quinclorac-resistant *E. crus-galli* compared with the susceptible biotype, which was consistent with the proteomic sequencing results. Overall, this study demonstrated that the resistant *E. crus-galli* enhanced its adaptability to quinclorac by improving the photosynthetic efficiency and GST activity.

## 1. Introduction

Barnyardgrass (*Echinochloa crus-galli*) is one of the top 15 herbicide-resistant weeds around the world which causes massive rice yield losses [1,2]. The use of different herbicides to control *E. crus-galli* in paddy fields has led to the rapid development of herbicide-resistant *E. crus-galli*. Currently, *E. crus-galli* shows resistance to several herbicides, such as penoxsulam [3], propanil [4], and clomazone [5]. Quinclorac (3,7-dichloro-quinoline-carboxylic acid) has, in the past two decades, been used as an excellent herbicide against paddy field *E. crus-galli*. Prolonged quinclorac exposure has caused herbicide-driven selective pressure [6], rapidly resulting in the development of resistance in *E. crus-galli* and exacerbating the loss of yield and rice quality [7]. Thus, understanding the molecular mechanism of *E. crus-galli* resistance to quinclorac is not only essential for controlling the development of their resistance but also for improving the rice yield and quality.

Currently, 513 unique cases of herbicide-resistant weeds, which includes 154 dicots and 113 monocots, are known to occur globally. Weeds have evolved resistance to 21 out of the 31 known herbicide sites of action, making them resistant to 165 different herbicides, and their resistance was reported in 96 crops in 71 countries [8]. Consequently, extensive efforts have been made to reveal their molecular mechanism of herbicide resistance. For example, a Trp-574-Leu mutation in the acetolactate synthases, *EcALS1* and *EcALS3,* of two high-level penoxsulam-resistant populations was detected in *E. crus-galli* [9]; in goosegrass (*Eleusine indica* (L.) Gaertn), a novel A212T mutation in the chloroplast protoporphyrinogen oxidase (*PPO1*) conferred resistance to the PPO inhibitor oxadiazon [10] and Pro-106-Ser amino acid substitution, while the heterozygous double TIPS (Thr-102-Ile + Pro-106-Ser) mutation (plus Pro-381-Leu) caused glyphosate resistance [11]. In annual bluegrass (*Poa annua* L.), the Ala-205-Phe substitution in the acetolactate synthase conferred resistance to imidazolinone, sulfonylurea, triazolopy rimidines, sulfonylamino-carbonyl-triazolinones, and pyrimidinyl (thio) benzoate herbicides [12]. In addition, the Ile-2041-Val mutation in the Acetyl-coenzyme A carboxylase (*ACCase*) gene could confer resistance to clodinafop-propargyl in the American sloughgrass (*Beckmannia syzigachne* Steud) [13]. Similarly, reduced herbicides absorption, their enhanced metabolism, and the detoxification capacity are also key factors contributing to the weed resistance to herbicides [14]. Moreover, the gene copy number variation and differences in the expression levels were shown to influence the evolution of herbicide resistance. For example, a cytochrome P450 (P450) gene, *CYP81A68*, is highly expressed in R than S plants, and its overexpression in rice seedlings caused tolerance to penoxsulam, cyhalofop-butyl, and metamifop [15]. The up-regulated ATP-binding cassette (ABC) transporter (EcABCC8) caused *Echinochloa colona* to exhibit resistance to glyphosate [16].

For quantitative proteomic studies, isobaric tags for a relative and absolute quantitation (iTRAQ) assay can allow the examination of multiple samples to be examined during a single mass spectrometry (MS) analysis, thus significantly reducing the experimental errors in individual experiments. In addition, iTRAQ-based approaches were used to examine the crop growth and development, as well as the molecular mechanisms associated with the environmental stresses responses in various plants, such as Wucai (*Brassica campestris* L.) [17], rice (*Oryza sativa* L.) [18], maize (*Zea mays* L.) [19], upland cotton (*Gossypium hirsutum* L.) [20], and soybean (*Glycine max* L.) [21]. The currently available genome sequences of weeds, such as barnyardgrass [22] and Chinese sprangletop (*Leptochloa chinensis*) [23], have provided vital references for studying their occurrence in crop fields and herbicide responses. Our previous study compared the adaptability of quinclorac resistance in the susceptible barnyardgrass under drought and salt stress [24,25]. As a continuation, the present study employed iTRAQ to identify proteins related to quinclorac resistance in barnyardgrass with an aim to provide insight into the proteomic changes associated with quinclorac treatment. This is a large-scale proteomic study to examine herbicide responses in different *E. crus-galli* biotypes. Through digging relevant information of proteins, genes, and their pathways, the findings will enhance the understanding of ecological fitness and the molecular mechanism of herbicide resistance in *E. crus-galli*, thereby potentially benefiting future research efforts in weed science.

## 2. Materials and Methods

### 2.1. Plant Materials

The experiments were conducted at Hunan Agricultural Biotechnology Research Institute, Changsha, Hunan. The seeds of quinclorac resistance (R) and the susceptible (S) *E. crus-galli* were collected from two adjacent rice fields in Chunhua, Hunan Province, in 2013. Compared with the S biotype with a GR_50_ of 22.38 g. a.i./ha, the GR_50_ of R (3106.94 g. a.i./ha) was much higher than the recommended field dose (225–375 g. a.i./ha) in China. Based on the resistance index, R had 138.86-fold higher resistance than S [25]. After determining the resistance level, resistant and susceptible biotypes were, respectively, propagated for four generations at breeding pools from 2014 to 2017. Briefly, seeds were sterilized, germinated, and grown in a climate-controlled chamber. After three weeks, the seedlings (with 4–5 leaves) were exogenously treated with 2.153 g/L quinclorac (Jiangsu Tianrong Corporation, Nanjing, China) using a spray tower (3WP-2000, Nanjing, China). Approximately 72 h after spraying, both treated and untreated control seedlings were harvested in three independent biological replicates for each treatment, snap-frozen in liquid nitrogen, and stored at −80 °C until use. Untreated and quinclorac-treated S samples were tagged as SCK and Squin, while the untreated and quinclorac-treated R samples were tagged as RCK and Rquin, respectively.

### 2.2. Measurements of Photosynthetic Parameters and Physiological Indexes

R and S *E. crus-galli* biotypes were planted in plastic pots in a climate-controlled chamber using the same conditions as described above. Leaf gas-exchange measurements were performed on the first fully expanded leaf of each plant using a Li-6400XT portable photosynthesis system (Li-Cor Inc., Lincoln, NE, USA) to determine the net photosynthetic rate (*Pn*), stomatal conductance (*Gs*), intercellular CO_2_ concentration (*Ci*), and transient transpiration rate (*Tr*), following a previously reported method with few modifications (Zhang et al., 2008) [26]. The reference concentration of ambient CO_2_ was about 370 parts per million (ppm) and the temperature was maintained at 26 °C. Measurements were made at a photosynthetic photon flux density of 1000 μmol m^−2^ s^−1^, as provided by a Q-Beam light source. All measurements were conducted between 10:00 a.m. and 12:00 p.m./noon on a sunny day.

Total Chl was measured by method of Arnon [27] with slight modification. A total of 1.0 g (W) fresh leaf was ground in ethanol (95%, 3 mL): SiO_2_: CaCO_3_ at certain mass ratio, then added 95% ethanol to 25 mL (V), and placed at approximately 4 °C for 24 h for chlorophyll extraction under dark conditions. The absorbance was determined with a UV–vis spectrophotometer (A560, AOE, Shanghai, China) at wavelengths at 645 and 663 nm. The concentrations were calculated as (mg·g^−1^): C_Chla_ = (12.7 × A_663_ − 2.69 × A_645_) × V/(1000 × W); C_Chlb_ = 24.96 × A_649_ − 7.32 × A_665._

### 2.3. GST Activity of E. crus-galli

The samples were collected at 0, 1, 3, 5, 7, and 14 days after quinclorac treatment. The fresh leaves were removed and weighted, then wrapped with silver paper and cooled with liquid nitrogen before stored at −80 °C. Sample extracts were prepared by homogenizing leaf tissue (three-leaf stage) in liquid nitrogen. The homogenate was then centrifuged for 20 min at 12,000× *g* then filtered. A GST assay kit (Nanjing Jiancheng Bioengineering Institute, Nanjing, China, A004) was used, and protein content was determined by Bradford [28] method. Glutathione (GSH) formation was recorded at 28 °C with a spectrophotometer 721 (INESA, Shanghai, China) at 412 nm.

### 2.4. Protein Extraction, Digestion, and iTRAQ Labeling

Treated and untreated leaves were selected for total protein extraction in three biological replicates as described by Isaacson [29] and Yang [30], with few modifications (see Appendix A). Protein digestion was carried out with filter-aided sample preparation (FASP) procedure [31], which uses 8-plex iTRAQ reagents to label the resulting peptide mixture. Samples were tagged as follows: RCK and SCK were labeled with tags 113 and 114 while Rquin and Squin were labeled with tags 115 and 116, respectively.

### 2.5. Peptide Fractionation and Quantitative Proteomic Analysis Using Liquid Chromatography–Tandem Mass Spectrometry (LC-MS/MS)

To generate strong cationic exchange chromatography using an Agilent 1100 HPLC Purifier system (Agilent Technologies Inc., Santa Clara, CA, USA), the iTRAQ-labeled peptide mixtures were reconstituted and acidified in buffer A [acetonitrile (ACN)—H_2_O (2:98, *v*/*v*)] and loaded onto a 2.1 × 150 mm Agilent Zorbax Extend-C18 column (5 μm). The peptides were eluted at a flow rate of 300 μL min^−1^ with a linear gradient of buffer B [(ACN—H_2_O (90:10, *v*/*v*))] for 7 min, and gradient elution as follows: 98% buffer A for 8 min, 95–98% buffer A for 2 min, and 95–97% buffer A for 30 min, 60–75% buffer A for 12 min, 10% buffer A for 10 min, then maintained in 98% buffer A for 5 min. Elution was monitored by absorbance at 210 and 280 nm, and the fractions were collected every minute. The collected fractions were desalted using a C18 column, then vacuum dried for further LC-MS/MS analysis.

All analyses were performed using a Q-Exactive mass spectrometer (Thermo, Waltham, MA, USA) equipped with a Nanospray Flex source (Thermo, Waltham, MA, USA). Samples were loaded with a capillary C18 trap column (3 cm × 100 µm, 3 µm, 150 Å) then separated using a C18 column (15 cm × 75 µm, 3 µm, 120 Å, ChromXP Eksigent) on an EASY-nLC^TM^ 1200 system (Thermo, Waltham, MA, USA). The flow rate was 300 nL/min and linear gradient of 74–95% buffer A for 48 min, 62–74% buffer A for 13 min, 15% buffer A for 6 min, then finally maintained in 95% buffer A for 3 min (mobile phase A = ACN-H_2_O-FA (2:98:0.1, *v*/*v*/*v*) and B = ACN-H_2_O-FA (95:5:0.1, *v*/*v*/*v*)). Full MS scans were acquired in the mass range of 300–1600 m/z with a mass resolution of 70,000, and the automatic gain control (AGC) target value was set at 1,000,000. The twelve most intense peaks in the MS were fragmented with higher-energy collisional dissociation (HCD) with collision energy of 30 eV. MS/MS spectra were obtained with a resolution of 17,500 and a max injection time of 50 ms. The Q-E dynamic exclusion was set for 15 s and run under positive mode.

### 2.6. Database Searches and Bioinformatics Analysis

Proteome Discoverer TM 2.2 (Thermo Scientific, Waltham, MA, USA) software was used for iTRAQ protein identification and quantification. Raw data files were obtained from the *Ecrus-galli_v6*.prot reference database, then run against the Swiss-prot database (accessed on 5 January 2019), and 545,388 entries were detected. The search results were filtered with the criteria: significance threshold *p* < 0.05 (with 95% confidence) and ion score or expected cutoff <0.05 (with 95% confidence). The main parameters were as follows. MS tolerance: 20 ppm; MS/MS tolerance: 0.5 Da; enzyme: Trypsin; database: *Ecrus-galli_v6*.prot; fixed modification: Carbamidomethyl (C), iTRAQ-8-plex (N-term), iTRAQ-8-plex (K); variable modification: Oxidation (M), Acetyl (protein N-term); decoy database pattern: reverse; peptide FDR: 0.01; protein FDR: 0.01.

For protein quantification, the protein ratios were calculated as the median of only the unique peptides of the protein, and all peptide ratios were normalized by the median protein ratio. Differentially expressed proteins (DEPs) were assessed for significant up- or down-regulation. One-sample t test was used to detect significant differences between two developmental stages at *p* < 0.05. Proteins with a fold change of >1.2 or <0.833 were considered to be differentially expressed as the average ratios of 113/114 and 115/116.

### 2.7. Protein Annotation and Classification

The proteins were screened against the proteins in the Swiss-Prot and Omics Bean databases with Score Sequest HT > 0 and peptide ≥ 1 threshold. The results were subsequently imported into protein2GO for gene ontology (GO) annotation using GoPipe, a standalone package that integrates BLAST and InterProScan results [32], and proteins were categorized into the functional groups, including cellular component, biological process, and molecular function. The identified DEPs were then assigned to Kyoto Encyclopedia of Genes and Genomes (KEGG) pathways [33,34]. These analyses were collectively performed to determine the functional subcategories and metabolic pathways that were significantly enriched with DEPs.

### 2.8. Statistical Analysis

Data were expressed as mean ± SD with at least three biological replicates. The difference was analyzed with DPS V9 and means were compared by Ducan’s test at significance level of 0.05. The related figures were drawn using Sigma Plot 10.0.

## 3. Results

### 3.1. Photosynthetic Parameters and Physiological Index Measurements

The pigment levels and gas exchange parameters were measured. As a result, R biotypes of the non-quinclorac-treated *E. crus-galli* showed reductions in the Chl *a*, Chl *b*, and total Chl contents by 18.17%, 22.99%, and 18.77%, respectively, relative to the S biotypes (Figure 1a–c). An analysis of the photosynthesis parameters revealed that *Pn* and *Ci* exhibited a slight decrease of 12.84% and 6.69% in the R biotypes (12.711 ± 0.174 μmol CO_2_ m^−2^ S^−1^ and 212.593 ± 0.937 μmol CO_2_ mol^−1^) relative to the S biotypes (14.583 ± 0.162 μmol CO_2_ m^−2^ S^−1^ and 227.823 ± 0.577 μmol CO_2_ mol^−1^), respectively (Figure 2a,c). Similarly, *Tr* and *Gs* also showed a significant decrease in the R biotypes (1.051 ± 0.007 mmol H_2_O m^−2^ S^−1^ and 0.081 ± 0.001 mol H_2_O m^−2^ S^−1^) relative to the S biotypes (1.548 ± 0.004 mmol H_2_O m^−2^ S^−1^ and 0.119 ± 0.001 mol H_2_O m^−2^ S^−1^) (Figure 2b,d). In contrast, the quinclorac treatment significantly increased the Chl *b* content by 82.15% in the R biotypes relative to the S biotypes (Figure 3b), while the Chl *a* and total Chl content increased slightly in the R biotypes compared to the S biotypes (Figure 3a,c). When examining the various parameters relating to photosynthesis, the *Pn* and *Ci* slightly increased by 11.15% and 11.77% in the R biotypes (17.476 ± 0.118 μmol CO_2_ m^−2^ S^−1^ and 262.509 ± 0.797 μmol CO_2_ mol^−1^) relative to the S biotypes (15.723 ± 0.177 μmol CO_2_ m^−2^ S^−1^ and 234.873 ± 0.553 μmol CO_2_ mol^−1^) while treated with quinclorac (Figure 4a,c). In addition, *Tr* and *Gs* were significantly increased in the R biotypes (2.241 ± 0.018 mmol H_2_O m^−2^ S^−1^ and 0.150 ± 0.001 mol H_2_O m^−2^ S^−1^) relative to the S biotypes (1.799 ± 0.016 mmol H_2_O m^−2^ S^−1^ and 0.101 ± 0.001 mol H_2_O m^−2^ S^−1^) in the quinclorac-treated samples (Figure 4b,d).

### 3.2. GST Activity Analysis

In the present study, the enzymatic activity was determined using a GST assay kit, based on a spectrophotometer with the general GST substrate CDNB. As a typical GST substrate, the CDNB was used to detect the GST in different organisms [35]. The results showed that both the R and S biotypes GST activity rise sharply, induced by the quinclorac within a certain period of time, and showed a trend of increasing first and then decreasing. When treated for 3 days, the GST activity of the R biotypes reached 445.55 U/mgprot, which was significantly lower than that of S (724.47 U/mgprot). The GST activity of the S biotypes was restored to the normal level at the 5th day after the quinclorac treatment, while the R biotypes were still significantly higher than the normal value at the 7th day. Until the 14th day, the activity of the GST in R was restored to the normal level. While without quinclorac treatment, the GST activity of R and S biotypes was relatively stable at different times, and the GST activity of R was significantly lower than that of S (Figure 5).

### 3.3. Identification of Proteins in E. crus-galli Using iTRAQ

The total proteins in the leaves of *E. crus-galli* were extracted from the quinclorac-treated and non-treated R and S biotypes. A bioinformatic analysis identified a total of 1076 DEPs from the 1814 unique peptides (score sequence HT > 0 and unique peptides ≥ 1). Proteins with a fold change (FC) > 1.2 (*p* < 0.05) or an FC < 0.833 (*p* < 0.05) were considered up- or down-accumulated, respectively. In addition, a total of 945 DEPs (including 131 identical differentially expressed proteins between RCK vs. SCK and Rquin vs. Squin) were identified from the 1814 unique proteins (Figure 6). For a functional prediction, all the quantified proteins were searched through the Omics Bean database and GO annotation database, and approximately 1433 GO annotated terms were assigned to proteins in three groups, including the biological process, cellular component, and molecular function using the BLAST 2GO pipeline.

### 3.4. Differentially Accumulated Proteins Species between the Susceptible (SCK) and Resistant (RCK) E. crus-galli Samples

Comparative proteomics between the RCK and SCK biotypes were conducted to identify the DEPs according to the aforementioned parameters. As a result, 46 up-accumulated and 194 down-accumulated proteins were detected in the RCK samples at *p* < 0.05, relative to the SCK samples (Figure 7 and Appendix A). The enrichment analysis assigned 41 down-accumulated proteins from the RCK samples to 160 GO terms, while 174 up-accumulated proteins were assigned to 564 GO terms (Appendix A). Compared to the susceptible *E. crus-galli*, differential proteins in the resistant biotypes mainly participated in biological processes, such as the metabolic process, cellular process, and response to stimulus. In addition, their roles in the cellular component, such as the cell, membrane, and extracellular region, as well as their roles in the molecular function, such as the binding and catalytic activity, were detected. The pathway enrichment detected the “energy metabolism”, “carbohydrate metabolism”, “signal transduction”, and “translation” as the key enrichment pathways (Figure 8). The top enriched pathways included the “ribosome (KO03010)”, “phenylpropanoid biosynthesis (KO00940)”, “pyruvate metabolism (KO00620)”, “carbon fixation in photosynthetic organisms (KO00710)”, “purine metabolism (KO00230)”, and “photosynthesis (KO03010 and KO00196)” (Appendix A). Within the “ribosome” category, nine proteins were enriched, and seven ribosomal proteins were up-accumulated, while only two (EC_v6.g028872.t1 and EC_v6.g106454.t1) were down-accumulated (Appendix A). A total of 4 photosynthesis-related and 4 environmental stimulus response protein species were down-accumulated in the RCK samples (Table 1).

### 3.5. Identification of DEPs between the Susceptible (Squin) Samples and Resistant (Rquin) E. crus-galli after Quinclorac Treatment

A total of 836 DEPs were identified between the Rquin and Squin *E. crus-galli* samples after the quinclorac treatment, with 428 and 408 showing up-accumulated and down-accumulated patterns, respectively (Appendix A). Compared with the control group (RCK vs. SCK), the number of DEPs in the resistant *E. crus-galli* were significantly increased after the treatment. Of the 836 DEPs, 756 were annotated into GO categories and KEGG pathways. In the Rquin samples, 376 proteins were up-accumulated and assigned to 663 GO terms, while 380 proteins with down-accumulated patterns were assigned to 852 GO terms (Figure 9 and Appendix A). The pathway enrichment analysis detected that “energy metabolism”, “carbohydrate metabolism”, “folding, sorting and degradation”, “transport and catabolism”, and “amino acid metabolism” were the main enrichment pathways (Figure 10). The most highly enriched pathways included the “carbon fixation in photosynthetic organisms (KO00710)”, “glycolysis/Gluconeogenesis (KO00010)”, “glyoxylate and dicarboxylate metabolism (KO00630)”, “photosynthesis (KO00195 and KO00196)”, “pyruvate metabolism (KO00620)”, “ribosome (KO03010)”, and “peroxisome (KO04146)” pathways (Appendix A). A total of 40 photosynthesis-related protein species were up-accumulated in the Rquin samples, while 8 herbicide-resistant-related proteins, such as the Glycosyltransferases (GT), GST, protoporphyrinogen oxidase (PPO), ABC transporter (ATP-binding cassette transporter), pyruvate dehydrogenase (PDH), were up-accumulated (Table 2). Other enriched categories containing up-accumulated protein species included “peroxisome”, “glutathione metabolism”, “methane metabolism”, and “starch and sucrose metabolism”.

## 4. Discussion

Proteins are the basis of life, with specific biological activities. In this study, a proteomic analysis was used to reveal the differences between the quinclorac-susceptible and -resistant *Ecrus-galli* biotypes at the protein level and to determine the ecological adaptability of their resistance.

### 4.1. Comparative Proteomics Reveal DEPs between Susceptible and Resistant Biotypes of E. crus-galli

Proteomics is an efficient method for studying complicated protein species in many numerous organisms. iTRAQ is a reliable peptide-labeling technique for quantifying proteins, which can detect more differentially abundant proteins than the traditional 2D gel-based methods and has enabled the accurate comparison of expressed proteins during drought, cold, and other stress responses [30,36,37]. This study performed an iTRAQ-based comparative proteomics analysis between the quinclorac-susceptible and -resistant *E. crus-galli*. As a result, numerous accumulated proteins that included functionally characterized and uncharacterized proteins were identified (Appendix A). A total of 46 up-accumulated and 194 down-accumulated DEPs were detected between the quinclorac-resistant and -susceptible *E. crus-galli* biotypes, while the quinclorac treatment generated 428 up-accumulated and 408 down-accumulated proteins between the resistant and susceptible biotypes, indicating a significant increase compared with the untreated group. Our extensive proteomic data provide valuable information for not only clarifying the differential accumulation patterns of proteins in the quinclorac-resistant and the susceptible biotypes but also facilitating the determination of the molecular mechanism underlying the herbicide resistance in *E. crus-galli*. However, the functions of these proteins still remain uncertain, thus warranting further analysis.

### 4.2. Ecological Fitness Costs Associated with Quinclorac Resistance of E. crus-galli

Leaf is the main site of photosynthesis and is necessary for plant growth and bioenergy synthesis [38,39]. Photosynthesis is an extremely complex process that involves photosynthetic pigments, reaction center complexes, and an electron transport system. During photosynthesis, the light-dependent reactions require four major protein complexes, photosystem I (PSI), PSII, cytochrome b6f complex, and ATP synthase. The PSII and PSI reaction centers, where photosynthetic charge separation occurs, consist of several protein subunits and cofactors, such as chlorophylls, carotenoids, and quinones [40]. Light-harvesting antenna complexes are coupled with the core complexes of both PSs to transfer the absorbed light energy to the reaction centers. In green algae and land plants, transmembrane proteins belonging to the light-harvesting chlorophyll *a*/*b*-binding protein family form the light-harvesting complex (LHC) for the PSI (LHCI) and PSII (LHCII) with chlorophylls and carotenoids [41]. LHC-II accounts for about one-third of the total thylakoid membranes protein, and accumulates about half the total chlorophyll in plants [42], with the LHC b content having a direct correlation with the Chl content in *Arabidopsis* [43]. In the present study, several proteins associated with photosynthesis were down-accumulated in the R biotype relative to the S biotype, which was consistent with the Chl *a*, Chl *b*, Chl *(a + b)* (Figure 1), and *Pn* (Figure 1 and Figure 2a). These down-accumulated proteins and lower photosynthetic parameters in the R biotype suggest that herbicide resistance in the R biotype leads to a reduction in the photosynthetic capacity, thus lowering the overall fitness. The quinclorac treatment of the Rquin biotype leads to an overaccumulation of photosynthesis-related proteins. This is in agreement with the opinion [44] that there may be a regulatory mechanism in the R biotype allowing the expression of related quick recovery genes which in turn prevent severe quinclorac damage to the plants. Previous reports have demonstrated that photosynthesis decreases during drought [45,46], cold [47], salt [48], and other abiotic stresses [49]. Similarly, an iTRAQ-based approach was used to show the down-accumulation of photosynthetic components in the multi-herbicide-resistant *E. crus-galli* biotype relative to the susceptible biotype [50].

Numerous plants have evolved an induced resistance, which is acquired under stress for survival, and the cost of fitness can arise from both internal and external plant processes [51], or from ecological interactions [51]. In this study, the pigment quantification and photosynthetic parameters suggested that the quinclorac-resistant R biotype had reduced the photosynthetic capacity, which lowered its overall fitness. These results are consistent with previous studies that showed a reduced photosynthetic capacity, photosynthetic potential, growth rates, resource competitive ability, and sexual reproduction in the R biotype [52].

### 4.3. Key Differentially Expressed Herbicide-Related Proteins Involved in the E. crus-galli Resistance to Quinclorac

The accumulation of a herbicide-related protein, such as acetolactate synthase (ALS) and acetyl-coenzyme A carboxylase (ACCase), was not detected in the R biotype in this study, which was in contrast to a previous report [49]. However, the contents of photosynthesis-related proteins, such as the oxygen-evolving enhancer protein and PS I reaction center subunit, as well as the response to environmental stimuli-related proteins, such as the Chlorophyll *a*-*b*-binding protein, were all down-accumulated in the resistant biotype (Table 1), which was consistent with a previous result that quinclorac-resistant *E. crus-galli* is less adaptable to the environment than the sensitive biotype [53,54]. Quinclorac treatment up-accumulated the contents of herbicide-related proteins, such as the protoporphyrinogen oxidase (PPO), glycosyltransferase (GT), and ABC transporter, but down-accumulated those of the GST and cytochrome P450 (CYP450) in the R biotype (Table 2). We speculate that the large increase in the GST activity in both the R and S biotypes from Day 1 to Day 3 after the quinclorac treatment (Figure 5) resulted from the detoxification effect, and a sharp decrease in the GST activity after Day 5 might be ultimately caused by herbicide damage. In plants, it was demonstrated that GSTs can detoxify specific herbicides [55,56,57]. The increase degree of the GST activity in the R biotype was lower than that of S, indicating that GST-mediated non-target-site-based resistance (NTSR) was likely to exist in this particular quinclorac-resistant *E. crus-galli*. In addition, photosynthesis and environmental stimulus response-related proteins were all up-accumulated, which indicated that the resistant *E. crus-galli* was more adaptable to quinclorac stress than the sensitive biotype. Moreover, a previous study demonstrated that the cytochrome P450s-mediated enhanced rate of the mesosulfuron-methyl metabolism was potentially involved in NTSR, and three key proteins annotated as esterase, GST, and glucosyltransferase were identified, which could potentially be used as transcriptional markers for the rapid diagnosis of metabolic mesosulfuron resistance in *Alopecurus aequalis* species [58].

### 4.4. Plant Defense Involved in Quinclorac-Resistant E. crus-galli

Quinclorac is a specific auxin herbicide playing roles in the chloro-quinolinecarboxylic acid structure and ethylene biosynthesis, and inducing plant necrosis and death [59,60]. Herbicide stress can induce changes in the abscisic acid (ABA) content in plants, for example, while being treated by quinclorac, the salicylic acid (SA) treatment down-regulated ABA genes more in rice (cultivar XS 134), which correlated with the enhanced tolerance to quinclorac-induced oxidative stress in the cultivar rice [61], and dicamba (3,6-dichloro-2-methoxybenzoic acid) can induce the ethylene accumulation in plants, which in turn inhibits plant growth by enhancing the ABA content [62]. Notably, no difference was observed in the quinclorac absorption, conduction, and metabolism in the resistant and susceptible *E. crus-galli* after treatment. However, quinclorac increased the ABA content in the dicotyledons of the susceptible plants, which is consistent with the reported physiological and biochemical characterization of quinclorac resistance in *Galium spurium* [63]. In the present study, the abscisic stress-ripening proteins (EC_v6.g060659.t1 and EC_v6.g005261.t1) were found to be down-accumulated in the R biotype relative to the S biotype after treating the quinclorac, indicating that the resistant *E. crus-galli* was more adaptable to quinclorac stress than the sensitive biotype.

## 5. Conclusions

In conclusion, we found quinclorac resistance in *E. crus-galli* and endow a fitness cost to the environment, and 1076 proteins were identified in the quinclorac-resistant and -susceptible *E. crus-galli* by a comparative proteomics analysis using iTRAQ. Several pathways potentially associated with photosynthesis and herbicide resistance were revealed. A measurement of the physiological and biochemical indexes combined with a comparative proteomics analysis demonstrated the cost of fitness for the environmental adaption of quinclorac-resistant *E. crus-galli*.

Based on the current resistance level and ecological adaptability of quinclorac-resistant *E. crus-galli*, it is difficult to control this type of weed and extend the useful life of quinclorac. In the future, we will discover and analyze the functions of the key genes related to the photosynthesis and GST in resistant *E. crus-galli*. These further functional studies are needed to examine the potential ecological fitness costs and the mechanisms of *E. crus-galli* for quinclorac resistance.

## Figures and Tables

**Figure 1 plants-12-00696-f001:**
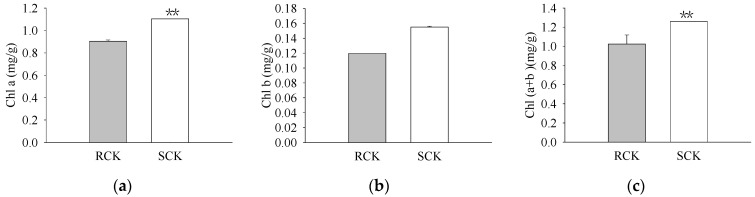
The chlorophyll (Chl) content of quinclorac-resistant (RCK) and -sensitive (SCK) *E. crus-galli* without quinclorac treatment. (**a**) Chl *a* content; (**b**) Chl *b* content; (**c**) Chl (*a* + *b*) content. ** represents a statistically significant difference (*p* < 0.01).

**Figure 2 plants-12-00696-f002:**
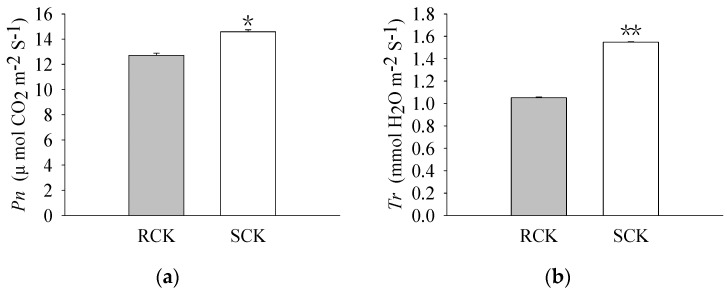
The photosynthetic parameters of quinclorac-resistant (RCK) and -sensitive (SCK) *E. crus-galli* without quinclorac treatment. (**a**) Net photosynthetic rate (*Pn*); (**b**) transpiration rate (*Tr*); (**c**) intercellular CO_2_ concentration (*Ci*); and (**d**) stomatal conductance (*Gs*). * represents a statistically significant difference (*p* < 0.05), and ** represents a statistically significant difference (*p* < 0.01).

**Figure 3 plants-12-00696-f003:**
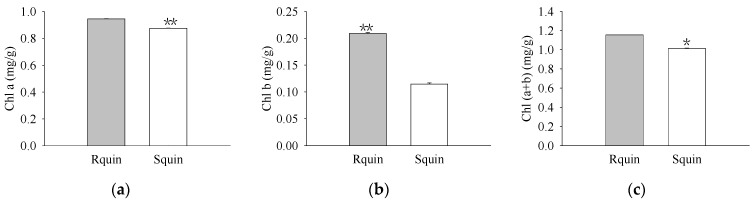
The chlorophyll (Chl) content of quinclorac-resistant (Rquin) and -sensitive (Squin) *E. crus-galli* with quinclorac treatment. (**a**) Chl *a* content; (**b**) Chl *b* content; (**c**) Chl (*a* + *b*) content. * represents a statistically significant difference (*p* < 0.05), and ** represents a statistically significant difference (*p* < 0.01).

**Figure 4 plants-12-00696-f004:**
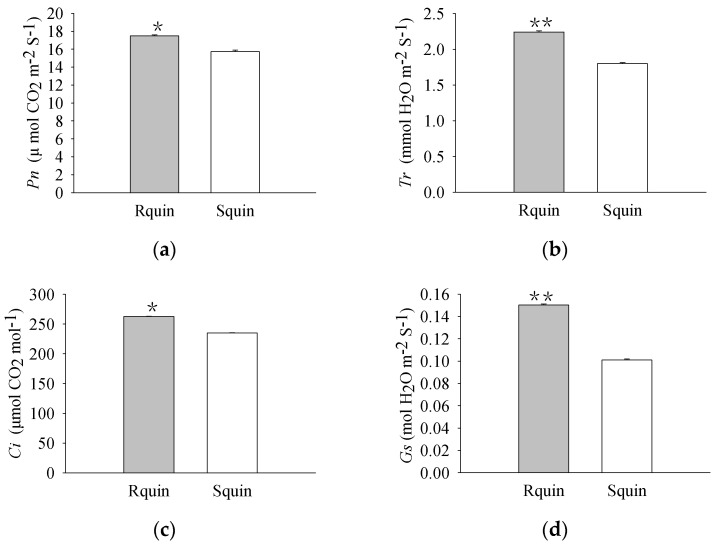
The photosynthetic parameters of quinclorac-resistant (Rquin) and -sensitive (Squin) *E. crus-galli* with quinclorac treatment. (**a**) Net photosynthetic rate (*Pn*); (**b**) transpiration rate (*Tr*); (**c**) intercellular CO_2_ concentration (*Ci*); and (**d**) stomatal conductance (*Gs*). * represents a statistically significant difference (*p* < 0.05), and ** represents a statistically significant difference (*p* < 0.01).

**Figure 5 plants-12-00696-f005:**
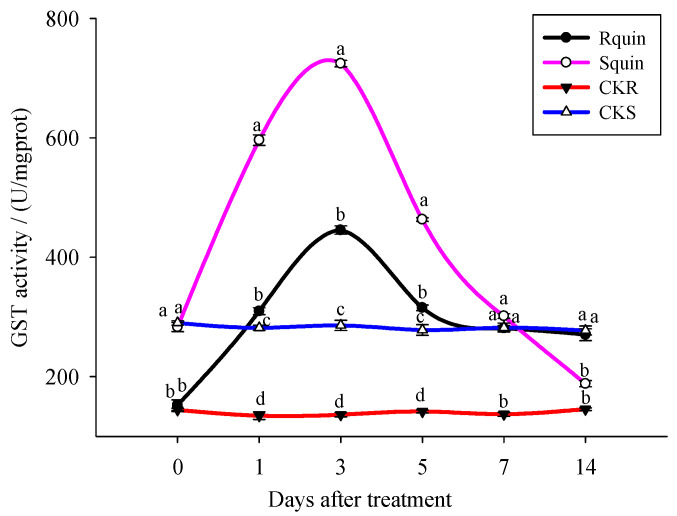
The GST activity of *E. crus-galli* under quinclorac treatment. Rquin: treating quinclorac-resistant *E. crus-galli* with quinclorac; Squin: treating quinclorac-sensitive *E. crus-galli* with quinclorac; CKR: treating quinclorac-resistant *E. crus-galli* with water; CKS: treating quinclorac-sensitive *E. crus-galli* with water. Lower case letters on histogram is expressed as least square means ± standard error (*p* < 0.05). Data are least square means ± standard error.

**Figure 6 plants-12-00696-f006:**
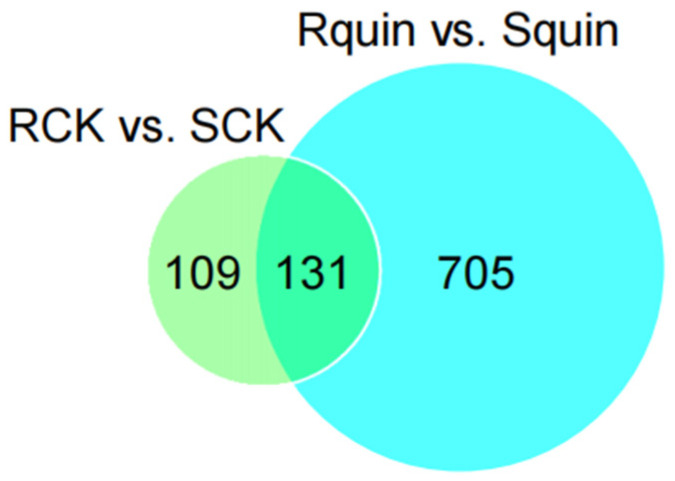
Quantitative analysis of the Venn analysis between RCK vs. SCK and Rquin vs. Squin.

**Figure 7 plants-12-00696-f007:**
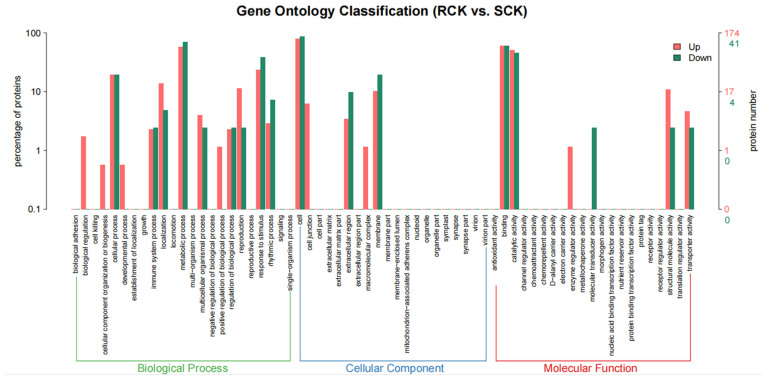
GO enrichment analysis map of differentially accumulated protein species between quinclorac-resistant and -sensitive *E. crus-galli* (RCK vs. SCK).

**Figure 8 plants-12-00696-f008:**
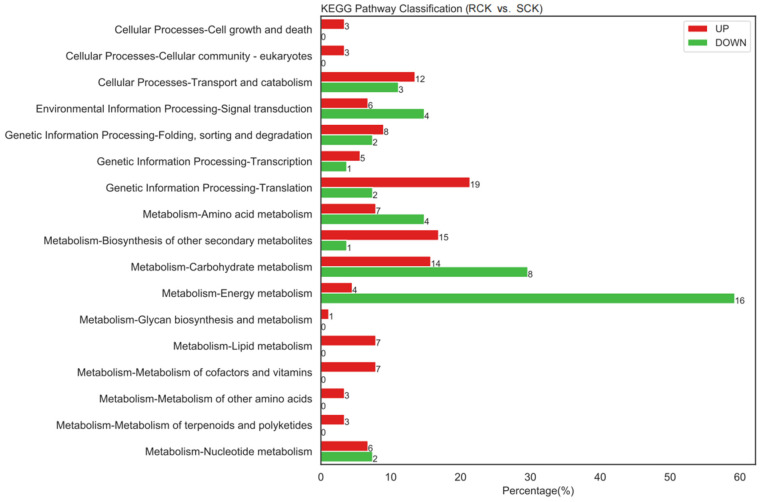
KEGG pathway enrichment analysis results between quinclorac-resistant (RCK) and -sensitive (SCK) *E. crus-galli*.

**Figure 9 plants-12-00696-f009:**
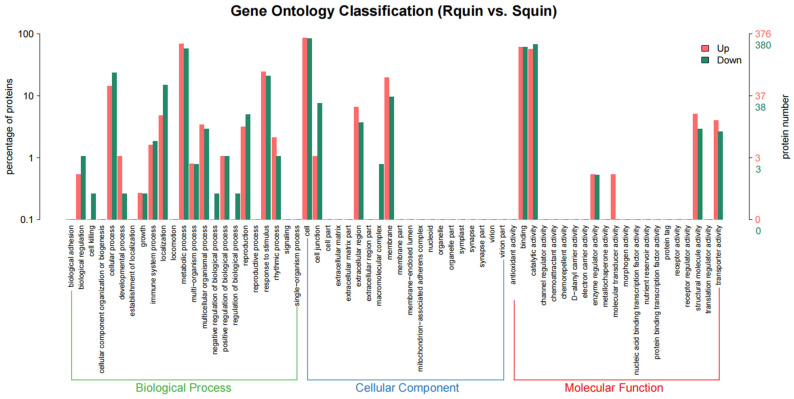
A GO enrichment analysis map of differentially accumulated protein species between quinclorac-resistant and -sensitive *E. crus-galli* with quinclorac treatment (Rquin vs. Squin).

**Figure 10 plants-12-00696-f010:**
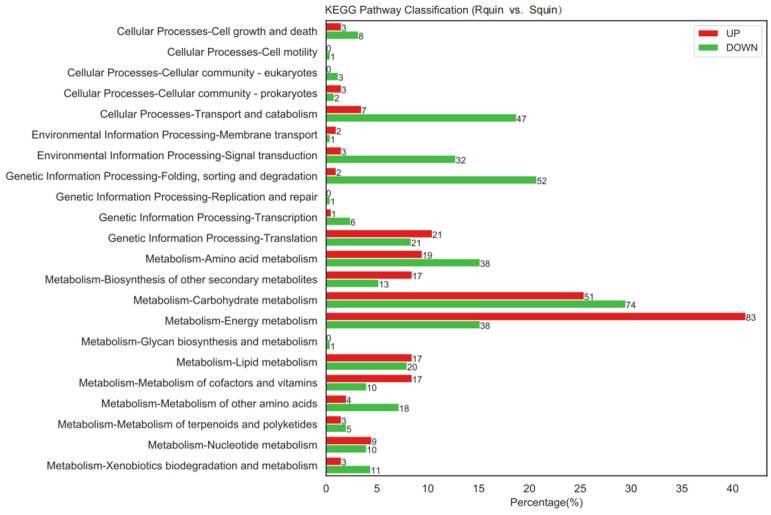
KKEGG pathway enrichment analysis results between quinclorac-resistant (Rquin) and -sensitive (Squin) *E. crus-galli* with quinclorac treatment.

**Table 1 plants-12-00696-t001:** The major differentially accumulated proteins between quinclorac-resistant and -sensitive (RCK vs. SCK) *E. crus-galli*.

Protein Annotation	Protein	Ratio (RCK/SCK)	*p* Value (<0.05)
Photosynthesis related			
↓ Oxygen-evolving enhancer protein 3-1	EC_v6.g069711.t1	0.791	2.08 × 10^−6^
↓ Oxygen-evolving enhancer protein 1	EC_v6.g003176.t1	0.811	0.005587
↓ Photosystem I reaction center subunit	EC_v6.g044816.t1	0.689	0.003137
↓ Photosystem I reaction center subunit III	EC_v6.g045919.t1	0.764	1.41 × 10^−5^
Environmental stimulus response			
↑ Linoleate 9S-lipoxygenase 4	EC_v6.g034212.t1	1.253	0.0303560
↑ Lipoxygenase 2.1	EC_v6.g007566.t1	1.686	1.53 × 10^−6^
↓ Chlorophyll *a*-*b*-binding protein CP24	EC_v6.g000874.t1	0.777	0.000563
↓ Chlorophyll *a*-*b*-binding protein	EC_v6.g025637.t1	0.746	4.27 × 10^−5^
↓ Chlorophyll *a*-*b*-binding protein 8	EC_v6.g020029.t1	0.826	6.50 × 10^−5^
↓ Chlorophyll *a*-*b*-binding protein 1	EC_v6.g011301.t1	0.800	7.06 × 10^−7^

↑: up-accumulated protein species; ↓: down-accumulated protein species.

**Table 2 plants-12-00696-t002:** The major differentially accumulated proteins between quinclorac-resistant (Rquin) and -sensitive (Squin) *E. crus-galli* with quinclorac treatment.

Protein Annotation	Protein	Ratio (RCK/SCK)	*p* Value (<0.05)
Photosynthesis related			
↑ Oxygen-evolving enhancer protein 3-1	EC_v6.g025748.t1	1.981	5.58 × 10^−7^
↑ Oxygen-evolving enhancer protein 1	EC_v6.g003176.t1	1.693	0.000172
↑ Oxygen-evolving enhancer protein 2	EC_v6.g084339.t1	1.689	2.31 × 10^−6^
↑ Photosystem I reaction center subunit VI	EC_v6.g077844.t1	1.829	0.001093
↑ Photosystem II stability/assembly factor	EC_v6.g057479.t1	1.577	3.57 × 10^−6^
↑ Photosystem II repair protein PSB27-H1	EC_v6.g090372.t1	1.660	3.7 × 10^−7^
↑ Photosystem I reaction center subunit psaK	EC_v6.g095902.t1	1.623	0.00021
↑ Photosystem II 22 kDa protein	EC_v6.g011916.t1	1.620	1.07 × 10^−6^
↑ Phosphoglycerate kinase	EC_v6.g045755.t1	1.411	1.73 × 10^−6^
↑ Photosystem I reaction center subunit II	EC_v6.g031932.t1	1.703	0.000355
↑ Photosystem I reaction center subunit N	EC_v6.g001270.t1	1.338	0.00048
↑ Photosystem I reaction center subunit VI	EC_v6.g077844.t1	1.829	0.001093
↑ Photosystem I reaction center subunit III	EC_v6.g031714.t1	1.525	5.34 × 10^−7^
↑ Photosystem II 10 kDa polypeptide	EC_v6.g005861.t1	1.501	8.89 × 10^−7^
Environmental stimulus response			
↑ Lipoxygenase 2.3	EC_v6.g043701.t1	2.277	0.000153
↑ Lipoxygenase 2.1	EC_v6.g007566.t1	1.902	9.62 × 10^−6^
↑ Chlorophyll *a*-*b*-binding protein	EC_v6.g018199.t1	1.707	1.11 × 10^−6^
↑ Chlorophyll *a*-*b*-binding protein 1	EC_v6.g011301.t1	1.567	1.4 × 10^−7^
↑ Chlorophyll *a*-*b*-binding protein CP24 10A	EC_v6.g000874.t1	1.536	0.000411
↑ 10B Chlorophyll *a*-*b*-binding protein CP24 10B	EC_v6.g078443.t1	1.932	3.12 × 10^−7^
↑ Chlorophyll *a*-*b*-binding protein 8	EC_v6.g107213.t1	1.551	1.16 × 10^−5^
↑ Chlorophyll *a*-*b*-binding protein 8	EC_v6.g020029.t1	1.684	2.67 × 10^−6^
↑ Chlorophyll *a*-*b*-binding protein CP24 10A	EC_v6.g023390.t1	1.617	9.02 × 10^−6^
↑ Chlorophyll *a*-*b*-binding protein of LHCII	EC_v6.g025637.t1	1.307	0.000138
↑ Chlorophyll *a*-*b*-binding protein	EC_v6.g052592.t1	1.252	0.004444
↑ Light-harvesting complex-like protein 3	EC_v6.g087330.t1	1.338	0.000663
↑ Chlorophyll *a*-*b*-binding protein	EC_v6.g095902.t1	1.623	0.00021
↑ Chlorophyll *a*-*b*-binding protein CP26	EC_v6.g052598.t1	1.573	2.36 × 10^−5^
↑ Chlorophyll *a*-*b*-binding protein CP29.2	EC_v6.g025652.t1	1.460	2.51 × 10^−6^
↑ Chlorophyll *a*-*b*-binding protein 1	EC_v6.g067789.t1	1.957	3.84 × 10^−7^
↑ Zeaxanthin epoxidase	EC_v6.g107058.t1	1.371	0.000274
↑ Glutamine synthetase	EC_v6.g047795.t1	1.365	0.000343
↑ Soluble inorganic pyrophosphatase 6	EC_v6.g072964.t1	1.284	0.000202
↑ Soluble inorganic pyrophosphatase	EC_v6.g042273.t1	1.468	2.34 × 10^−5^
↑ Glyceraldehyde-3-phosphate dehydrogenase (GAPB)	EC_v6.g096773.t1	1.279	1.04 × 10^−5^
↑ Glyceraldehyde-3-phosphate dehydrogenase (GAPB)	EC_v6.g052452.t1	1.332	0.000154
↑ Glyceraldehyde-3-phosphate dehydrogenase (GAPB)	EC_v6.g098973.t1	1.252	0.001558
↑ Glyceraldehyde-3-phosphate dehydrogenase A	EC_v6.g023371.t1	1.269	1.77 × 10^−6^
Abscisic acid regulation			
↑ chaperonin	EC_v6.g095910.t1	1.391	0.000658
↓ Abscisic stress-ripening protein 5	EC_v6.g060659.t1	0.669	0.001379
↓ Abscisic stress-ripening protein 3	EC_v6.g005261.t1	0.409	0.008321
Ribosome related			
↑ Ribosomal protein L5	EC_v6.g096733.t1	1.890	1.06 × 10^−5^
↑ Ribosomal protein L12-2	EC_v6.g107141.t1	2.049	1.80 × 10^−7^
↑ Ribosomal protein S9	EC_v6.g041745.t1	1.773	0.000391
↑ Ribosomal protein 2	EC_v6.g094158.t1	1.754	0.002046
↑ Ribosomal protein L3	EC_v6.g069774.t1	1.616	4.27 × 10^−5^
↑ Ribosomal protein L15	EC_v6.g027262.t1	1.407	0.014191
↑ Ribosomal protein L4	EC_v6.g059034.t1	1.868	6.74 × 10^−5^
Herbicide resistance related			
↑ Protoporphyrinogen oxidase	EC_v6.g076423.t1	1.361	7.52 × 10^−5^
↑ Glycosyltransferase family 64	EC_v6.g029743.t1	1.478	0.001409
↓ Glutathione S-transferase 4	EC_v6.g031522.t1	0.682	0.000671
↓ Glutathione S-transferase GST 23	EC_v6.g083691.t1	0.507	0.000455
↑ Glutathione S-transferase GSTU1-like	EC_v6.g031756.t1	1.409	0.002851
↓ Glutathione S-transferase U8	EC_v6.g028330.t1	0.602	2.43 × 10^−5^
↓ Glutathione S-transferase GSTF1	EC_v6.g021913.t1	0.642	0.037605
↓ Glutathione S-transferase Z2	EC_v6.g011773.t1	0.674	3.48 × 10^−6^
↑ Glutathione S-transferase F11	EC_v6.g048450.t1	1.201	0.000632
↓ Glutathione S-transferase	EC_v6.g001351.t1	0.627	0.018221
↓ Glutathione reductase	EC_v6.g080772.t1	0.744	0.002028
↓ Glutathione reductase	EC_v6.g040387.t1	0.580	0.000583
↑ ABC transporter B family member 28	EC_v6.g079748.t1	1.321	0.002159
↑ ABC transporter I family member 6	EC_v6.g046423.t1	1.294	0.017675
↑ Glutathione S-transferase F11	EC_v6.g048450.t1	1.201	0.000632
↓ Cytochrome P450	EC_v6.g010860.t1	0.435	6.93 × 10^−7^
↓ Cytochrome P450	EC_v6.g010858.t1	0.677	0.03581
Defense response			
↑ Nodulin-related protein	EC_v6.g047444.t1	1.594	0.000101
↑ Glucan endo-1,3-beta-glucosidase	EC_v6.g083656.t1	1.411	1.9 × 10^−5^
↑ Patatin-like protein	EC_v6.g067097.t1	1.345	0.000146
↑ Pathogenesis-related protein	EC_v6.g068270.t1	1.310	0.00011
↑ Prohibitin-2	EC_v6.g042981.t1	1.236	0.001687
Oxidation-reduction related			
↑ Peroxidase 70	EC_v6.g085666.t1	2.016	0.000473
↑ Peroxidase 4	EC_v6.g048714.t1	1.518	4.11 × 10^−5^
↑ Peroxidase 52	EC_v6.g080097.t1	1.319	0.024639
↑ Fructose-1,6-bisphosphatase	EC_v6.g022027.t1	1.249	0.002185

↑: up-accumulated protein species; ↓: down-accumulated protein species.

## Data Availability

Not applicable.

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
