# Peer review of "Proteomic Analysis Comparison on the Ecological Adaptability of Quinclorac-Resistant Echinochloa crus-galli"

_plants, 2023, doi:10.3390/plants12040696_

Round 1
Reviewer 1 Report
The work contains very extensive material. A number of more or less advanced and complicated quantitative and qualitative analyzes were performed in the study. However, in my opinion, the obtained results do not confirm the assumptions contained in the title of the manuscript. The authors themselves in the conclusions include information that the assessment of ecological effects and understanding the mechanisms requires further research.
However, the main criticism of the work concerns the description of the acquisition and selection of plant material. There is no information on the method of resistance assessment (e.g. bioassay and ED50 determination). We do not know what the resistance index was for resistant biotypes. Whether a material with a similar IR was selected for further research. Whether the type of resistance was previously proven (enzymatic or at the site of action - PCR analyses).
The results of analyzes related to the process of photosynthesis are consistent with most of the literature data. A certain novelty is the in-depth analysis of proteins.
In the title, instead of the word "reveals", "comparison" is more appropriate. The claim that such an analysis "reveals" ecological adaptation processes has not been proven. Perhaps it rather gives new possibilities and research directions for the evaluation of such processes.
In addition, there are many editorial errors in the work, e.g.
Line 136: cooed - cooled,
Line 139: centrifuged for 20 min at 12000 g then filtered
Line 135: The samples were collected at 0, 1, 3, 5, 7, and 14 days (after …..?)
Why is this research material published only 5 years after the analysis?
As I wrote above, the results of the work are not convincing for me and do not correspond to the title of the manuscript. However, I believe that after changing the title and supplementing the description in the part concerning plant material, the work could be published.
Unfortunately, the differences in the life processes of the R and S biotypes are not visible in real field conditions. The R and S biotypes are biometrically indistinguishable and reproduce similarly, which poses a problem for the farmer and makes it difficult to develop anti-immunity programmes.
Author Response
Jan 16, 2023
Dear Reviewer:
Thank you very much for your reviewers’ comments concerning our manuscript entitled ‘Proteomic analysis reveals the ecological adaptability of quinclorac-resistant Echinochloa crus-galli’ (Manuscript ID: plants-2147934). Those comments are all valuable and very helpful for revising and improving our paper, as well as the important guiding significance to our researches. We have studied your comments carefully and have made corrections which we hope meet with approval. We revised the title and other mistakes, and explained why not detailed description of plant materials in this paper. Revised portions are marked in blue in the paper. The main corrections in the paper and the responds to the reviewer’s comments are as flowing:
Question 1:The work contains very extensive material. A number of more or less advanced and complicated quantitative and qualitative analyzes were performed in the study. However, in my opinion, the obtained results do not confirm the assumptions contained in the title of the manuscript. The authors themselves in the conclusions include information that the assessment of ecological effects and understanding the mechanisms requires further research.
Answer: Thanks for your comments. Mechanism of resistance and ecological adaptability of E. crus-galli to quinclorac is complicated, our research aims to explore the ecological adaptability of quinclorac resistant E. crus-galli on the level of proteomics. In the futher, we will discover key genes related to photosynthesis and GST in resistant E. crus-galli by transcriptomics and analyze thier function. Of cause, further functional studies are needed to examine the potential ecological fitness costs and the mechanisms of E. crus-galli for quinclorac resistant.
And we will change the title as “Proteomic analysis comparison on the ecological adaptability of quinclorac-resistant Echinochloa crus-galli” in the new version.
Question 2:However, the main criticism of the work concerns the description of the acquisition and selection of plant material. There is no information on the method of resistance assessment (e.g. bioassay and ED50 determination). We do not know what the resistance index was for resistant biotypes. Whether a material with a similar IR was selected for further research. Whether the type of resistance was previously proven (enzymatic or at the site of action - PCR analyses).
Answer: Thanks for your comments. Plant materials in this research were collected in 2013 from two adjacent rice fields in Chunhua, Hunan Province, and relative resistance index (R/S) of resistance population is 138.86, and propagated by ourselves which mentioned in our previous studies(Line 99-106) (References 25 in this paper) .
Question 3:In the title, instead of the word "reveals", "comparison" is more appropriate. The claim that such an analysis "reveals" ecological adaptation processes has not been proven. Perhaps it rather gives new possibilities and research directions for the evaluation of such processes.
Answer: Thanks for your suggestion. The title revised as “Proteomic analysis comparison on the ecological adaptability of quinclorac-resistant Echinochloa crus-galli”.
Question 4:Line 136: cooed - cooled. Line 139: centrifuged for 20 min at 12000 g then filtered. Line 135: The samples were collected at 0, 1, 3, 5, 7, and 14 days (after …..?)
Answer: Thanks for your corrections. We revised these mistake: “cooed” revised as “cooled” (142), “The samples were collected at 0, 1, 3, 5, 7, and 14 days” revised as“ The samples were collected at 0, 1, 3, 5, 7, and 14 days after quinclorac treatment”. (Line 140-141).
Question 5:Why is this research material published only 5 years after the analysis?
Answer: This research is part of my doctoral dissertation, it takes nearly 3 years for analysis and related experiments, and then given birth to a child, you know is a hard work for caring a little baby and undertaking some other work simultaneously. So it takes a long time for this research.
As I wrote above, the results of the work are not convincing for me and do not correspond to the title of the manuscript. However, I believe that after changing the title and supplementing the description in the part concerning plant material, the work could be published.
Unfortunately, the differences in the life processes of the R and S biotypes are not visible in real field conditions. The R and S biotypes are biometrically indistinguishable and reproduce similarly, which poses a problem for the farmer and makes it difficult to develop anti-immunity programmes.
Answer: Thanks for your comments. The title had been changed as your suggestion. As for supplementing the description in the part concerning plant material, we quoted our previous research (reference 25), so we think is not necessary to repeat the plant materials particularly in this paper. Of cause, if you consider it is necessary, we will supplement it.
We appreciate for warm work of editor and reviewer earnestly, and hope that the correction will meet with approval.
Once again, thank you very much for your comments and suggestion, we are looking forward to hearing from you soon.
Sincerely yours,
Lamei Wu
Lamei Wu (corresponding author)
Hunan Agricultural Biotechnology Research Institute
Hunan Academy of Agricultural Sciences
Changsha, Hunan 410125, P. R. CHINA
E-mail: wlmlamei@hunaas.cn
Tel: 86-0731-84696075; Fax: 86-0731-84696075

Reviewer 2 Report
The authors have presented a manuscript, describing the proteomic response of Echinocloa crus-gally genotypes to herbicide resistant. Following, I have included some comments to improve the manuscript.
- I suggest to the authors to add a new section detailing the state of the art. In this section, authors have to describe the relevant related work in which explain.
- Can the authors include at the end of the introduction, more details of the objectives of their study.
- This work presents very interesting results and practice to response of Echinocloa genotypes. I think that the authors can improve the format of results demonstration. The authors can highlight better the importance of the results obtained.
- Conclusions. Consider extending the conclusions and adding a Future works paragraph. The summary and Conclusions, it is better to combine them in only section of Conclusions.
Finally, the review is interesting and presents considerable information on the possibilities of the resistant of Echinocloa genotypes, but authors must improve the presentation of their results and discussion. The topic in interesting, but the study lacks more details with precision, and concrete conclusion that will help farmers to improve or to change their strategies in the agriculture.
Author Response
Jan 16, 2023
Dear Reviewer:
Thank you very much for your reviewers’ comments concerning our manuscript entitled ‘Proteomic analysis reveals the ecological adaptability of quinclorac-resistant Echinochloa crus-galli’ (Manuscript ID: plants-2147934). Those comments are all valuable and very helpful for revising and improving our paper, as well as the important guiding significance to our researches. We have studied your comments carefully and have made corrections which we hope meet with approval. We added a new section detailing plant defense involved in quinclorac resistant E. crus-galli, and revised several details which could meet the suggestion. Revised portions are marked in blue in the paper. The main corrections in the paper and the responds to the reviewer’s comments are as flowing:
Question 1:I suggest to the authors to add a new section detailing the state of the art. In this section, authors have to describe the relevant related work in which explain.
Answer: Thanks for your suggestion, and this is an excellent point. We added a new section as “4.4. Plant defense involved in quinclorac resistant E. crus-galli”. In this section, we have replenished three relevant references (reference [59-61]), and describe the relevant related work about about our research, such as “When comparing the resistant and sensitive biotypes in the present study, the abscisic stress-ripening proteins (EC_v6.g060659.t1 and EC_v6.g005261.t1) were found to be differentially down-accumulated in the resistant biotype relative to the sensitive biotype after treated with quinclorac, one of the ABA relevant proteins (EC_v6.g005261.t1) was even further down-accumulated (2.4-fold) in the resistant biotype.”(Line 509-514).
Question 2:Can the authors include at the end of the introduction, more details of the objectives of their study.
Answer: Thanks for your suggestion. We have added details of the objectives of this study as “This is a large-scale proteomic study to examine herbicide stress responses in different barnyardgrass biotypes. Through digging relevant information of proteins, genes and pathway” (Line 93-95).
Question 3:This work presents very interesting results and practice to response of Echinocloa genotypes. I think that the authors can improve the format of results demonstration. The authors can highlight better the importance of the results obtained.
Answer: Thanks for your suggestion, that is an excellent point. We optimized the format of Fig 6 (Line 319 ), and we also highlight upregulated genes in bold red color and downregulated in bold blue color in Table 1 and Table 2 for easily identification. Additional format of data and figure changes would take more than a month, missing the due date for revision submission, so only minor changes have been made for the format of results.
Question 4:Conclusions. Consider extending the conclusions and adding a Future works paragraph. The summary and Conclusions, it is better to combine them in only section of Conclusions.
Answer: Thanks for your suggestion. We have extended the conclusions and added a future works paragraph. The section of conclusion is “In summary, here we show in E. crus-galli population that quinclorac resistance endows fitness cost to environment. 1076 proteins were identified in the quinclorac treated and non-treated resistant and susceptible E. crus-galli by comparative proteomics analysis using iTRAQ. Several pathways potentially associated with photosynthesis and herbicide resistance were revealed. Measurement of physiological and biochemical indexes combined with comparative proteomics analysis demonstrated the cost of fitness for the environmental adaption of quinclorac resistant E. crus-galli.
Based on the current resistance level and ecological adaptability of quinclorac resistant E. crus-galli, it is difficult for weed control and extending the life of quinclorac. In the futher, we will discover key genes related to photosynthesis and GST in resistant E. crus-galli by transcriptomics and analyze thier function. These further functional studies are needed to examine the potential ecological fitness costs and the mechanisms of E. crus-galli for quinclorac resistant”(Line 520-534).
Question 5:Finally, the review is interesting and presents considerable information on the possibilities of the resistant of Echinocloa genotypes, but authors must improve the presentation of their results and discussion. The topic in interesting, but the study lacks more details with precision, and concrete conclusion that will help farmers to improve or to change their strategies in the agriculture.
Answer: Thanks for your suggestion. We have added our speculation about the changes of GST activity in both R and S biotypes in the Discussion section as “We speculate that the large increase in GST activity in both R and S biotypes from Day 1 to Day 3 after quinclorac treatment (Fig. 5) was due to detoxification effect, and sharply decrease in GST activity from Day 5 to Day 14 might by the reason of ultimately herbicide damage. In plants, it has been demonstrated that GSTs can detoxify specific herbicides[55-57]. The increase degree of GST activity in R biotype was lower than that of S, indicating that GST-mediated non-target-site based resistance (NTSR) likely was present in this particular quinclroac resistant E. crus-galli, but maybe not the only main factor. (Line 477-486)”
We appreciate for warm work of editor and reviewer earnestly, and hope that the correction will meet with approval.
Once again, thank you very much for your comments and suggestion, we are looking forward to hearing from you soon.
Sincerely yours,
Lamei Wu
Lamei Wu (corresponding author)
Hunan Agricultural Biotechnology Research Institute
Hunan Academy of Agricultural Sciences
Changsha, Hunan 410125, P. R. CHINA
E-mail: wlmlamei@hunaas.cn
Tel: 86-0731-84696075; Fax: 86-0731-84696075

Reviewer 3 Report
Comments to manuscript „Proteomic analysis reveals the ecological adaptability of 2 quinclorac-resistant Echinochloa crus-galli”, submitted by Wu et al.
The manuscript presents an interesting study by investigating differences on the protein levels of quinclorac- resistant and susceptible E. crus-galli, when treated with the herbicide in comparison to the untreated controls. Methods for the construction of the proteome profiles are adequate and the experiments, including the GST activity and other measurements seem to be done carefully and obtained data robust.
In the text, there are only a few complains. In line 158, should it be: The peptides were….? Add the explanation of Rquin, Squin, CKR and CKS also in the legend of Figure 5. The legend of Figure 8 should be similar to the one of Figure 10.
Unfortunately, the results are only descriptive and the possible reasons for the observed differences in the proteins ‘up-or down accumulations are not discussed. Therefore, the study is incomplete, as the phenomenon cannot really explained. The key question, why the quinclorac treatment of the Rquin plants leads to overaccumulation, for instance of photosynthesis related proteins, is not answered, which demands other methods than only proteomic ones. As numerous, time-consuming experiments, necessary to get insights in the molecular background of the plant response, have to be performed, it is recommended to discuss at least some possibilities, which could be causative for the increased accumulation of related proteins. This would improve the manuscript.
Author Response
Jan 16, 2023
Dear Reviewer:
Thank you very much for your reviewers’ comments concerning our manuscript entitled ‘Proteomic analysis reveals the ecological adaptability of quinclorac-resistant Echinochloa crus-galli’ (Manuscript ID: plants-2147934). Those comments are all valuable and very helpful for revising and improving our paper, as well as the important guiding significance to our researches. We have studied your comments carefully and have made corrections which we hope meet with approval. We revised few descriptive mistakes, added the explanation in the legend of Figure 5. We also added the explain of the possible reasons for down-accumulated proteins in R biotype in the section of discussion. Revised portions are marked in blue in the paper. The main corrections in the paper and the responds to the reviewer’s comments are as flowing:
Question 1:In the text, there are only a few complains. In line 158, should it be: The peptides were….? Add the explanation of Rquin, Squin, CKR and CKS also in the legend of Figure 5. The legend of Figure 8 should be similar to the one of Figure 10.
Answer: Thanks for your comments. “The peptide was” was revised as “the peptides were” (Line 164). Add the explanation of Rquin, Squin, CKR and CKS in the legend of Figure 5 as “Rquin: quinclorac-resistant E. crusgalli treated with quinclorac; Squin: quinclorac-sensitive E. crusgalli treated with quinclorac; CKR: quinclorac-resistant E. crusgalli untreated with quinclorac; CKS: quinclorac-sensitive E. crusgalli untreated with quinclorac”(Line 300-303). The legend of Figure 8 was revised as “KEGG pathway enrichment analysis results between quinclorac-resistant (RCK) and -sensitive (SCK) E. crusgalli untreated with quinclorac” (Line 355-356).
Question 2:Unfortunately, the results are only descriptive and the possible reasons for the observed differences in the proteins ‘up-or down accumulations are not discussed. Therefore, the study is incomplete, as the phenomenon cannot really explained. The key question, why the quinclorac treatment of the Rquin plants leads to overaccumulation, for instance of photosynthesis related proteins, is not answered, which demands other methods than only proteomic ones. As numerous, time-consuming experiments, necessary to get insights in the molecular background of the plant response, have to be performed, it is recommended to discuss at least some possibilities, which could be causative for the increased accumulation of related proteins. This would improve the manuscript.
Answer: Thanks for your comments. We added the explain of the possible reasons for down-accumulated proteins in R biotype as “These down-accumulated proteins and lower photosynthetic parameters in R biotype suggest that herbicide resistance in the R biotype leads to a reduction in photosynthetic capacity, thus lowering the overall fitness. The quinclorac treatment of the Rquin biotype leads to overaccumulation of photosynthesis related proteins, we are agree with the opion[44] that there may be a regulatory mechanismin the R biotype that allowed the expression of related genes to be significantly unaffected or to rapidly recover, in turn preventing severe damage to the plants caused by quinclorac(Line 440-448)”.
We appreciate for warm work of editor and reviewer earnestly, and hope that the correction will meet with approval.
Once again, thank you very much for your comments and suggestion, we are looking forward to hearing from you soon.
Sincerely yours,
Lamei Wu
Lamei Wu (corresponding author)
Hunan Agricultural Biotechnology Research Institute
Hunan Academy of Agricultural Sciences
Changsha, Hunan 410125, P. R. CHINA
E-mail: wlmlamei@hunaas.cn
Tel: 86-0731-84696075; Fax: 86-0731-84696075

Reviewer 4 Report
Dear authors,
I think this manuscript is suitable for this journal, but all of your figures should be more clear. Please provide higher resolution.
Author Response
Jan 16, 2023
Dear Reviewer:
Thank you very much for your identification and reviewer’ comment concerning our manuscript entitled ‘Proteomic analysis reveals the ecological adaptability of quinclorac-resistant Echinochloa crus-galli’ (Manuscript ID: plants-2147934). Revised portions are marked in blue in the paper. The main corrections in the paper and the responds to the reviewer’s comments are as flowing:
Question:I think this manuscript is suitable for this journal, but all of your figures should be more clear. Please provide higher resolution.
Answer: Thanks for your comments. We modified the Fig.6 (Line 319), and submitted clear picture. Besides, other figures in this manuscript were submitted by EPS or PDF format.
Once again, thank you very much for your comments and suggestion, we are looking forward to hearing from you soon.
Sincerely yours,
Lamei Wu
Lamei Wu (corresponding author)
Hunan Agricultural Biotechnology Research Institute
Hunan Academy of Agricultural Sciences
Changsha, Hunan 410125, P. R. CHINA
E-mail: wlmlamei@hunaas.cn
Tel: 86-0731-84696075; Fax: 86-0731-84696075

Round 2
Reviewer 1 Report
Thank you for your comment and corrections. It is satisfying for me. The description of the ECHCG biotypes is actually included in publication 25. It is an open access journal, however, I think an additional 2-3 sentences on resistance would be better. Not every reader has to (wants) to look in other literature sources.
In conclusion: I accept the comment and the corrections made, but I am counting on additional information about resistance.
Author Response
Jan 22, 2023
Dear Reviewer:
I am pleased to receive your comments about our manuscript entitled ‘Proteomic analysis comparison on the ecological adaptability of quinclorac-resistant Echinochloa crus-galli’ (Manuscript ID: plants-2147934). We have added the description of quinclorac resistant Echinochloa crus-galli which we hope meet with approval. The main corrections in the paper and the responds to the reviewer’s comments are as flowing:
Question:Thank you for your comment and corrections. It is satisfying for me. The description of the ECHCG biotypes is actually included in publication 25. It is an open access journal, however, I think an additional 2-3 sentences on resistance would be better. Not every reader has to (wants) to look in other literature sources.
In conclusion: I accept the comment and the corrections made, but I am counting on additional information about resistance.
.Answer: Thanks for your suggestion. We added the description of quinclorac resistant Echinochloa crus-galli as “ Compared with the S biotype with a GR50 of 22.38 g a.i./ha, the GR50 of R (3106.94g. a.i./ha) was much higher than the recommended field dose (225–375 g. a.i./ha) in China. Based on the resistance index, R had 138.86-fold higher resistance than S”(Line 104-107).
Once again, thank you very much for your comments and suggestion, we are looking forward to hearing from you soon.
Sincerely yours,
Lamei Wu
Lamei Wu (corresponding author)
Hunan Agricultural Biotechnology Research Institute
Hunan Academy of Agricultural Sciences
Changsha, Hunan 410125, P. R. CHINA
E-mail: wlmlamei@hunaas.cn
Tel: 86-0731-84696075; Fax: 86-0731-84696075

Reviewer 3 Report
Dear authors,
Please correct the English particularly in the added and changed parts of the revision.
Author Response
Jan 22, 2023
Dear Reviewer:
I am pleased to receive your comments about our manuscript entitled ‘Proteomic analysis comparison on the ecological adaptability of quinclorac-resistant Echinochloa crus-galli’ (Manuscript ID: plants-2147934). We have modified some English expression particularly in the added and changed parts of the manuscript which we hope meet with approval. The main corrections in the paper and the responds to the reviewer’s comments are as flowing:
Question:Please correct the English particularly in the added and changed parts of the revision.
.Answer: Thanks for your suggestion. We modified some English expression particularly in the added and changed parts of the manuscript (Line 94-95, Line 109, Line 302-305, Line 358, Line 446-449, Line 479-486, Line 497-498, Line 511-513, Line 518-521, Line 526-530).
Once again, thank you very much for your comments and suggestion, we are looking forward to hearing from you soon.
Sincerely yours,
Lamei Wu
Lamei Wu (corresponding author)
Hunan Agricultural Biotechnology Research Institute
Hunan Academy of Agricultural Sciences
Changsha, Hunan 410125, P. R. CHINA
E-mail: wlmlamei@hunaas.cn
Tel: 86-0731-84696075; Fax: 86-0731-84696075
